# The effects of Swiss summer camp on the development of socio-emotional abilities in children

Yves Gerber[1], Edouard Gentaz[2], Jennifer Malsert [2,3]*

**1** IDEA Lab, Faculty of Psychology and Educational Sciences, University of Geneva, Geneva, Switzerland, **2** SensoriMotor, Affective and Social Development Laboratory, Faculty of Psychology and Educational Sciences, University of Geneva, Geneva, Switzerland, **3** Department of Special Educational Needs, University of Teacher Education of the State of Vaud, Lausanne, Switzerland

* Jennifer.malsert@unige.ch

**Data Availability Statement:** All relevant data are within the paper and its Supporting information files.

**Funding:** The author(s) received no specific funding for this work.

## Abstract

This quasi-experimental research explores the relationship between participation in two-week summer camps and changes in children's altruism and self-esteem. Data were collected from 256 children aged 6 to 16 years. A self-reported altruism scale, a self-evaluation questionnaire and a temperament measure (EAS) were administered on two occasions either two weeks apart during the summer holidays or in class before and after the autumn holidays. The responses of 145 children attending summer camps were compared with those of 111 pupils. A significant increase in the altruism score was found between the pre-test and post-test in the camp condition, but no change in the children's self-esteem was found with the entire sample. Exploratory analyses suggest variables that may be associated with more favourable participation in summer camps; certain dimensions of temperament are among them, as well as factors related to the camps themselves. Differences in the increase of altruism and self-esteem scores in summer camp were observed according to the identified child profiles. The limitations of this work are highlighted before proposing perspectives for future research.

## Introduction

The development of socio-emotional abilities can be defined as the ability of the child to form close and secure adult and peer relationships, experience, regulate and express emotions, pay attention, make good decisions in response to the social problems that they will encounter and engage in a large number of prosocial behaviors in socially appropriate ways and explore the environment and learn in the different contexts: family, school, extra-school [1–3].

There is a political and scientific consensus on the importance of socio-emotional abilities in the everyday life [4] and in the academic world at the beginning of school [5–7]. These socio-emotional abilities also have the advantage of being trainable and implemented in school context, which makes them a prime target for universal prevention (e.g. school bullying, school failure, youth violence) [8–12].

**Competing interests:** The authors have declared that no competing interests exist.

Researchers have also looked at the development of social-emotional abilities outside of school time in the context of leisure time [13]. Some leisure time activities fall within what Mahoney et al. [14] refer to as organised activities, i.e. who, where, what and how participation takes place is specified. They emphasise that organised activities are contexts in which children and young people have the opportunity to develop a wide range of skills. Sports activities, for example, are one such activity. A systematic review of the literature on the benefits of sport participation in children and adolescents found, among other things, benefits for self-esteem and social interactions [15]. The authors note that the social nature of team sports seems to play a particularly favourable role in these benefits. Another element that appears to be important for nurturing self-esteem and building social skills is sustained relationships with many adults [16]. There are other structured activity settings.

Children and young people operate in a range of important learning contexts, of which summer camps are one [17]. These places of socialisation and experimentation [18] correspond to the criteria mentioned above, a type of structured activity that involves interactions with peers and supportive adults. It is therefore relevant to look more specifically at the role of this context for the development of social-emotional skills in order to assess the extent to which they contribute to their growth. A body of research suggests that this is indeed the case.

In 2005, the American Camp Association (ACA) published an extensive survey of over 5,000 families. Participants aged 8 to 14, their parents and staff completed questionnaires on three occasions: at the beginning and end of the camp, and then six months later for follow-up. This was the first US-wide research project to evaluate the effect of summer camp participation [19]. The results of this research project suggest that summer camps have a beneficial effect on several measured constructs [20]. They show growth in four areas: positive identity, social skills, physical and cognitive skills, and positive values and spirituality. This is reflected in increased scores on scales measuring self-esteem, social skills, independence, leadership qualities and adventurousness. The results of this survey and its methodology are described in several articles [20–22]. Mishna et al. [23] also found a positive effect of summer camps on prosocial skills. Several studies also support a positive effect of summer camps on children's self-esteem [20, 23, 24]. Overall, research suggests that summer camps are positive developmental contexts for children and young people [25, 26], while noting that, despite having benefits for most children, summer camps cannot provide an optimal experience for all children at all times [25]. This is beyond the scope of this research, which is concerned with the effect of summer camps on typical children, but it should be mentioned that a number of studies also suggest benefits for children with atypical development or special conditions [for examples, see 27–29].

Holiday camps, defined as residential and structured extracurricular activities taking place over several days during the holiday periods, have been experiencing a decline in attendance, for some years [30], and there is some desire to revalue them. To our knowledge, there is no research on this topic in Switzerland, and the data highlighted in the literature come mainly from the United States with positive effects, despite rather small effect sizes [19].

The main aim of the current study was to assess whether participation in summer camps could promote school-aged children's social-emotional abilities, in particular altruism and self-esteem. The secondary aim was to identify whether some children benefited more from the summer camp experience than others, and whether what distinguished them were individual characteristics and/or characteristics related to camp participation. For example, research shows that temperament, in interaction with other variables, may contribute to individual differences in altruism [31] and self-esteem [32]. And, certain characteristics, such as older age or having been to camp repeatedly, appear to be associated with more favourable camp experiences [25]. The third aim was therefore to investigate which individual and camp-specific

characteristics were most likely to be favourable to the development of altruism and self-esteem in children and young people in summer camps. The two research questions of the study were: (1) To what extent participation in summer camps is associated with increased self-esteem and altruism in children and adolescents aged 6–16 years? (2) What factors may influence the development of self-esteem and altruism in children and adolescents aged 6 to 16 who attend summer camps?

## Method

### Participants

A total of 256 French-speaking children aged 6 to 16 years took part in this quasi-experimental design study (Table 1). There were 145 participants in the Swiss summer camp group. Their average age was 11.20 years (SD = 2.20 years). Among them, there were 76 girls and 67 boys. Two did not indicate their sex. In this group, 16 children mentioned participating rarely (14.2%), 61 occasionally (54%), 27 often (23.9%) and 9 very often (8%) in summer camps. Thirty-two participants did not give any indication of their attendance at summer camps. Ninety-one of these children were in primary school (3th to 8th Harmos) and 43 in secondary school (9th to 11th Harmos). Eleven did not give any indication of their class.

The control group consisted of 111 children with a mean age of 9.43 years (SD = 1.39 years). Among them, there were 61 girls and 49 boys. One did not indicate its sex. In this group, 18 children indicated that they rarely (31%), 18 occasionally (31%), 19 often (32.8%) and 3 very often (5.2%) participated in summer camps. Fifty-three did not answer this question. One hundred and ten of these children were enrolled in primary school and one reported being enrolled in secondary school.

### Material

The data for this study were collected using questionnaires: an altruism scale, a self-esteem scale and a temperament measure. Control questions were added to these three measures. The

**Table 1. Characteristics of the sample by groups (camp vs control).**

| | *Camp* condition (n = 145) | *Control* condition (n = 111) | Total (n = 256) |
|---|---|---|---|
| Sociodemographic variables | | | |
| Age* | 11.20 (2.20) | 9.43 (1.39) | 10.43 (2.08) |
| Gender | | | |
| boys | 67 | 49 | 116 |
| girls | 76 | 61 | 137 |
| no answer | 2 | 1 | 3 |
| Pre-test scores | | | |
| Altruism | 29.92 (12.09) | 32.26 (10.21) | 30.95 (11.34) |
| Self-esteem | 0.99 (4.81) | 1.97 (4.73) | 1.44 (4.79) |
| Emotionality | 2.62 (0.91) | 2.61 (0.80) | 2.61 (0.86) |
| Activity* | 3.39 (0.74) | 3.64 (0.75) | 3.50 (0.75) |
| Sociability* | 3.88 (0.87) | 4.15 (0.68) | 4.00 (0.80) |
| Shyness | 2.55 (0.77) | 2.61 (0.75) | 2.58 (0.76) |

*Note.*

* means that there is a significant difference between the two conditions with p < .05.

The comparison of the groups shows differences at the first measurement between the camp group and the control group on some characteristics (age, activity and sociability). The participants did not differ significantly on the other characteristics.

same questionnaires were used with the children in the camp group and the control group, except that the control questions were adapted to the context.

**Adapted self-reported altruism scale [33].** This 14-item scale measures altruistic intentions [34]. It is an adaptation of Rushton's Self Report Altruism Scale by Witt and Boleman in 2009 [33]. It was freely translated into French by us for this study. Each item corresponds to a situation (e.g. "I would give directions to someone I did not know") for which the subject is asked to indicate how often he would perform this behaviour on a five-point scale. The scale is numbered from 0 (never) to 4 (very often). The altruism score is obtained by summing the points of the fourteen items. It can be between 0 and 56. The higher the score, the higher the level of altruistic intention. This scale has good internal consistency ($\alpha$ = .861).

**Self-assessment questionnaire [35].** This three-dimensional scale is composed of 9 items that use social comparison to obtain an estimate of children's self-esteem based on self-evaluations [35]. For each item (e.g., "How smart do you think you are compared to children your age?"), the child is asked to rate himself or herself on a five-point scale ranging from "Much less than the others" (-2) to "Much more than the others" (+2). The overall score of the self-assessments is between -18 and +18. A negative score indicates that the child feels inferior to others in the different domains, while a positive score indicates that the child perceives him/herself as superior to others. A child who perceived him/herself to be as good as others in all domains would have an overall score of 0 on this measure. In their validation studies, Maintier and Alaphilippe [35] highlight a cognitive dimension ("intelligence" and "schoolwork"), a social dimension ("pleasure in reading", "relationships with friends", "relationships with adults" and "ease of saying things") and a bodily dimension ("drawing, music and handicrafts", "sport" and "beauty"). The confirmatory factor analysis that we conducted supports the structure proposed by the authors, but as the alpha coefficients were insufficient for the social ($\alpha$ = .405) and bodily ($\alpha$ = .486) dimensions, only the global score was used in this study ($\alpha$ = .697).

**French Emotionality Activity and Sociability Questionnaire (EAS) [36].** This four-dimensional scale (emotionality, activity, sociability and shyness) measures temperament through 20 items [36]. Emotionality refers to the emotional style and intensity of emotional reactions, activity reflects the frequency and intensity of motor responses, sociability refers to preferring to be in a group rather than alone, and shyness mainly denotes tense and inhibited behaviour with unfamiliar people [36, 37]. Responses to the items are given on a five-point agreement scale ranging from 1 (strongly disagree) to 5 (strongly agree). Each dimension is measured by five items such as "I cry easily" (emotionality), "I run from morning to night" (activity), "I like to be with others" (sociability) and "I am rather shy" (shyness). This questionnaire yields four composite scores between 1 and 5. The sum of the items in each dimension is divided by the number of items (i.e. five). The higher the score on each dimension, the higher the level of the subject on the dimension in question. The results of the confirmatory factor analysis led us not to use item 18 ("If I am alone, I feel isolated") in the calculation of the sociability score. The internal consistencies of the scales are globally acceptable: emotionality ($\alpha$ = .697), activity ($\alpha$ = .570), sociability ($\alpha$ = .710) and shyness ($\alpha$ = .617).

**Socio-demographic and control questions.** The questionnaire included questions on the following socio-demographic characteristics for all participants: age, date of birth, sex and class. Children in the Swiss camp group were asked three additional specific groups of questions. In the first, they were asked to indicate by ticking a box (true or false) what was true for them from a list of six items (e.g. 'I am attending a summer camp for the first time'). In the second, they had to select from a set of five themes the type of camp they had just attended (e.g. "nature camp"). They had the possibility to tick several boxes and to add an answer themselves under "Other". In the third, they were asked about the frequency of their participation in holiday camps ("Have you regularly participated in holiday camps?"). The answer options were

"No, this is my first camp" and "Yes". If they ticked this box, they were asked the frequency on a four-point scale from 'rarely (less than one camp per year)' to 'very often (five or more camps per year)'. The latter question was also asked of children in the control group with slightly different wording ("Do you attend holiday camps?"). They could also answer "No" or "Yes" with the same response patterns for frequency. The other specific supplementary questions for the control group were about the activities they had during the holidays with multiple choice answers ('Days in general') for the first question. They had five closed response options to tick (e.g. "At home with your parents") and one open option ("Other"). The second question was an open question ("Did you have any particular activities during this holiday? (Travel, leisure)").

## Procedure

**Recruitment.** For the experimental summer group, a meeting was organised with the managers of each of the five Swiss holiday camp organisations that agreed to take part in this research to explain the context of the project and the conduct of the assessments. The consent forms were signed during the information meetings with the parents, directly at the camp site or sent by e-mail or post. These forms were signed by the parents and only by the children from the age of 14.

For the control group, the recruitment of classes was done according to a standard procedure through the Department of Public Instruction (DIP), which was responsible for making the applications and selecting the classes. Consent forms were signed by the parents.

**Completion of questionnaires.** In the summer camps, the questionnaires were administered during the summer holidays in two measurement times: at the beginning (pre-test) and at the end (post-test) of the camp. The time between the two measurements varied between 10 and 19 days (M = 12.40, SD = 1.89) depending on each summer camp. The administration was carried out by experimenters, by the camp staff or by both. Some camps preferred to administer the questionnaires themselves. In this case, they received a protocol with standardised instructions. They included explanations of how to complete the questionnaire and answers to questions about the vocabulary of the items.

For the completion of the questionnaires, the participants were seated at a table in a quiet area. Small groups were formed according to the age of the participants and their approximate reading level. There were mainly four standardised instructions that were given to the participants: (1) there is no right or wrong answer, the focus is on the child's opinion, (2) it is important to answer all questions and to tick only one box per question, (3) the questionnaire is to be filled in individually and (4) the questionnaire is anonymous. The surveys lasted between 20 minutes and 1 hour. At the end of the second measurement, a young scientist certificate was given to each participant.

In schools, the questionnaires were administered before and after the October holidays in three schools. The time between the two measurement times was between 11 and 16 days (M = 12.30, SD = 1.29). The questionnaires were filled in directly by the pupils in the classroom with the same instructions as during the camps. Two experimenters were present in each class to supervise the questionnaires and answer any questions of understanding. The tests lasted between 30 minutes and 1 hour. At the end of the two administrations, a young scientist certificate was given to each participant.

## Data analyses

**Altruism and self-esteem gain scores as function of groups.** In order to compare the 2 groups and measure the effect of summer camps, we calculated gain scores (G) (T1 = pre-test

scores, T2 = post-test scores, G = T2—T1) for each participant. We then performed an independent sample t-test on the gain scores and calculated an effect size for each result. When groups are not randomized and the two groups cannot be assumed to be from the same population, that is, they have the same mean apart from the variables included in the model (the so-called 'strongly ignorable treatment assignment' assumption [38–40]), this is a more robust method than ANCOVA. Two paired-sample t-tests were also conducted to determine whether altruism and self-esteem scores increased significantly between the two measurement times in each condition. The assumptions of the statistical analyses were checked. In case of outliers or violation of the normality assumption, the nonparametric alternatives proposed by Broc et al. [41] were also performed. In this article, the results of the parametric analyses, which are relatively robust [42, 43] and lead to the same conclusions, are reported. Missing data were excluded analysis by analysis. For all analyses, the significance level chosen was α = .05.

**Two complementary analyses: Random forest and hierarchical clustering.** The data was then explored using two types of analyses. As our interest is in the variables that influence the development of altruism and self-esteem of children in summer camp, these two analyses are conducted only on the data of the camp group.

The first was a random forest analysis with the *randomForest* package in RStudio to construct a hierarchy of variables based on the quantification of the importance of the effects on the variables of interest (i.e. the altruism and self-esteem gain scores). The index obtained allows the variables to be ranked from the most important to the least important without worrying about the form of the relationship between the variables [see 44]. The index used (% IncMSE) reflects the increase in the mean squared error (MSE) in the prediction of the model when the data for a predictor variable is randomly shuffled while maintaining the initial data for the other variables [45]. Therefore, the higher this index is, the more predictive the variable is for the model. This type of analysis, based on data mining, is already used in many fields and is gaining importance in psychological research [46].

The second type of exploratory analysis consisted of a hierarchical clustering, which is a type of cluster analysis that allows for the grouping of individuals on the basis of common characteristics in order to identify profiles [41, 47, 48]. This analysis was conducted for participants in the camp condition with the aim of identifying possible profiles of participants who would have shown greater or lesser changes in altruism and self-esteem scores. One-sample t-tests were conducted for each profile to determine whether the gain scores for altruism and self-esteem are significantly different from zero. The purpose of this exploratory section is to highlight avenues for future research. These results should be taken with caution as they are exploratory, especially as they require databases without missing values.

## Results

The characteristics of the participants in each group are summarised in Table 1.

### Altruism and self-esteem changes as function of groups

Analyses of the altruism gain scores show that the scores of children who attended summer camps increased significantly when compared to those of children in the control group ($t_{218}$ = 4.17, p < .001, d = .57) (Table 2). The group explained 7.39% of the variance in altruism gain scores.

More precisely, for the altruism score, the results of the paired-samples t-test in the camp condition indicate that the mean of the second measurement time (M = 31.83, SD = 11.53) is significantly higher than in the first measurement time (M = 29.78, SD = 12.24) (Fig 1A). The altruism score increased between the two measurements ($t_{125}$ = 3.63, p < .001). The effect size

**Table 2. Mean gain scores (and standard deviations) of dependent variables, the significance of statistical tests, and the effect size.**

| Variable | Mean gain scores (SD) | | T tests on gain scores | |
|---|---|---|---|---|
| | Camp | Control | t(p) | Effect size (d) |
| Altruism | 2.06 (6.36) | -1.54 (6.29) | 4.17 (< .001) | 0.57 |
| Self-esteem | -0.05 (4.10) | -0.02 (2.80) | -0.06 (.95) | n.s. |

is small (d = .32). In the control condition, the results indicate a decrease in the altruism score between the two measurement times ($t_{93}$ = -2.38, p = .02). The effect size is small (d = -.25). On average, the altruism score at the first measurement time (M = 32.44, SD = 10.08) is higher than at the second measurement time (M = 30.89, SD = 11.77).

Analyses of the self-esteem gain scores showed no significant difference in gain scores between children in the two conditions ($t_{226}$ = -0.06, p = .95) (Table 2). The proportion of variance in self-esteem gain scores explained by group is null.

In the camp condition, the results of the paired-samples t-test indicate that the mean of the second measurement time (M = 0.92, SD = 5.06) is not significantly higher than that of the first measurement time (M = 0.97, SD = 4.85) (Fig 1B). On average, the results do not indicate a change in the self-esteem score between the two sessions ($t_{122}$ = -0.13, p = .90). In the control condition, the results indicate a certain stability of the self-esteem score between the two measurement times ($t_{104}$ = -0.07, p = .95). On average, the esteem score at the first measurement time (M = 2.00, SD = 4.80) is similar to that obtained at the second measurement time (M = 1.98, SD = 5.24).

## Two complementary analyses: Random forest and hierarchical clustering

The random forest analysis with the *randomForest* package show that the variables that contribute most to prediction of the altruism gain score in summer camp are the camp organization (%IncMSE = 5.40), and the altruism (%IncMSE = 3.10) and the shyness (%IncMSE = 1.94) scores in the first measurement time. The variable that contributes most to the prediction of the summer camp self-esteem gain score is the self-esteem score at the first measurement time (%IncMSE = 3.13). The importance of the variables in predicting altruism and self-esteem gain scores is presented in Figs 2 and 3.

The hierarchical cluster analysis revealed four distinct profiles (Table 3). The categorical variables that contributed significantly to the constitution of the groups were the frequency of participation in holiday camps (p < .001), whether or not the child had gone to camp with friends (p < .001), whether or not the child had the opportunity to choose activities during

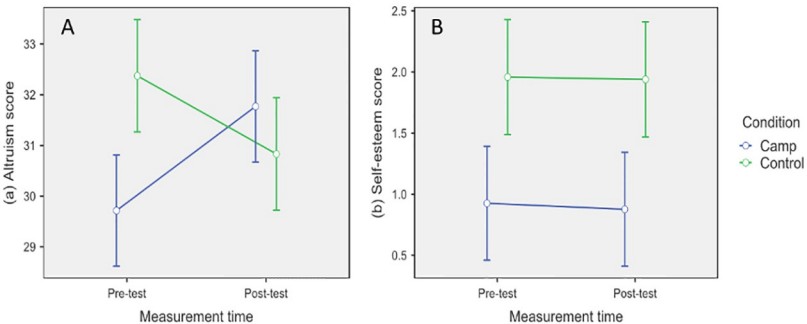

**Fig 1. Altruism (A) and self-esteem (B) scores with standard error for pre-test and post-test in the camp and control conditions.**

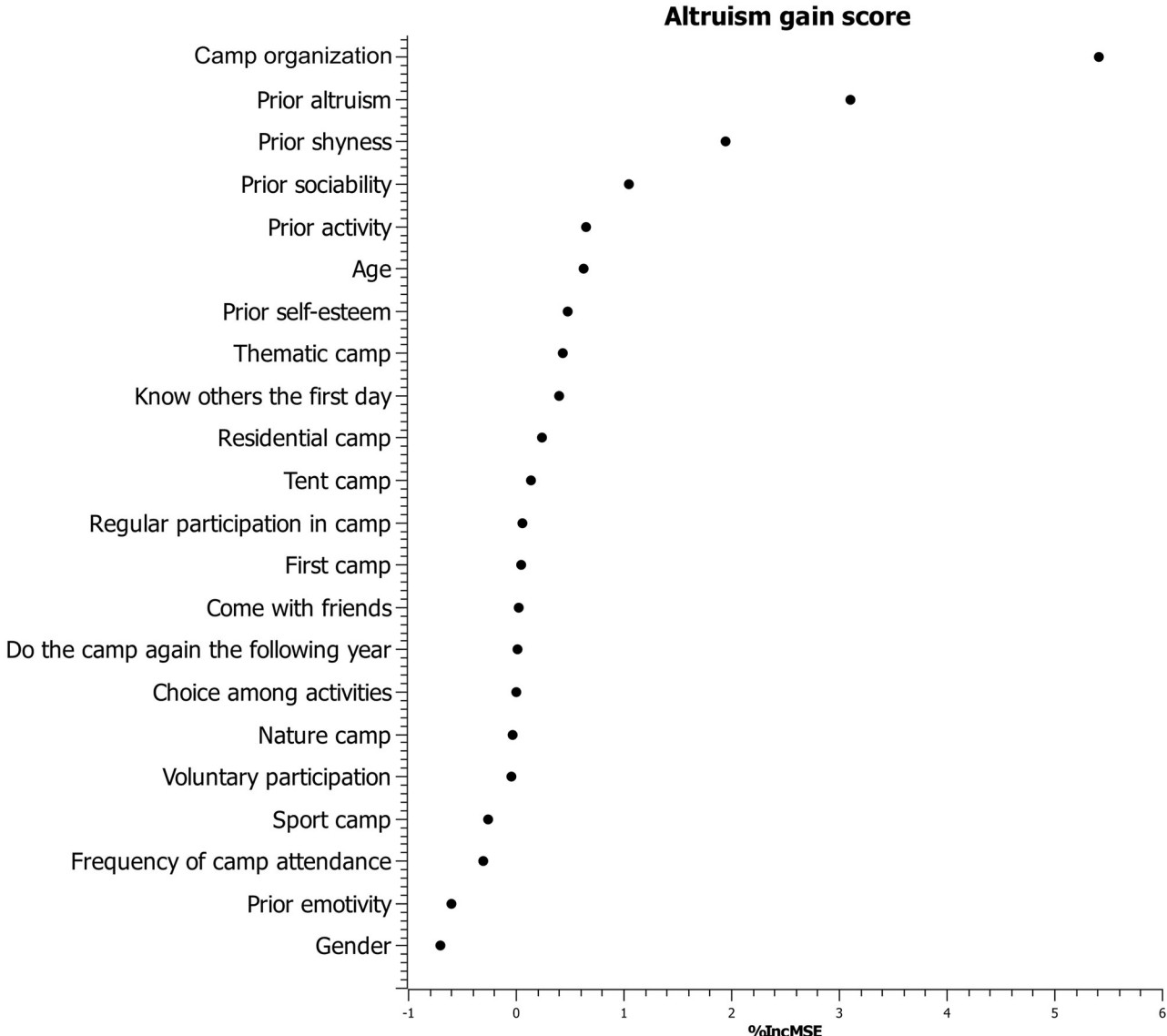

**Fig 2. Variables importance plot for altruism gain score in the camp condition.**

camp (p < .001), the camp organisation with which the child went to camp (p < .001), whether or not the child wanted to go to camp again the following year (p = .002), whether or not the child chose to go to camp (p = .006), and whether or not the child knew other children at the beginning of camp (p = .008). As for the quantitative variables that contributed significantly to the creation of the four profiles, these were age ($\eta 2 = 0.24$, p < .001), pre-test scores for altruism ($\eta 2 = 0.38$, p < .001), self-esteem ($\eta 2 = 0.16$, p = .001), activity ($\eta 2 = 0.14$, p = .003) and sociability ($\eta 2 = 0.11$, p = .014), and the altruism gain score ($\eta 2 = 0.13$, p = .005). As the description of the identified profiles indicates, it seems that the first (for altruism) and the second (for self-esteem) are the most favourable.

**First profile.** The first profile is composed of eighteen children who are on average younger than all the camp participants (M = 9.83 years, SD = 1.69 years) with a lower altruism score (M = 20.89, SD = 10.39) and a higher self-esteem score (M = 3.94, SD = 4.67) at the pre-test.

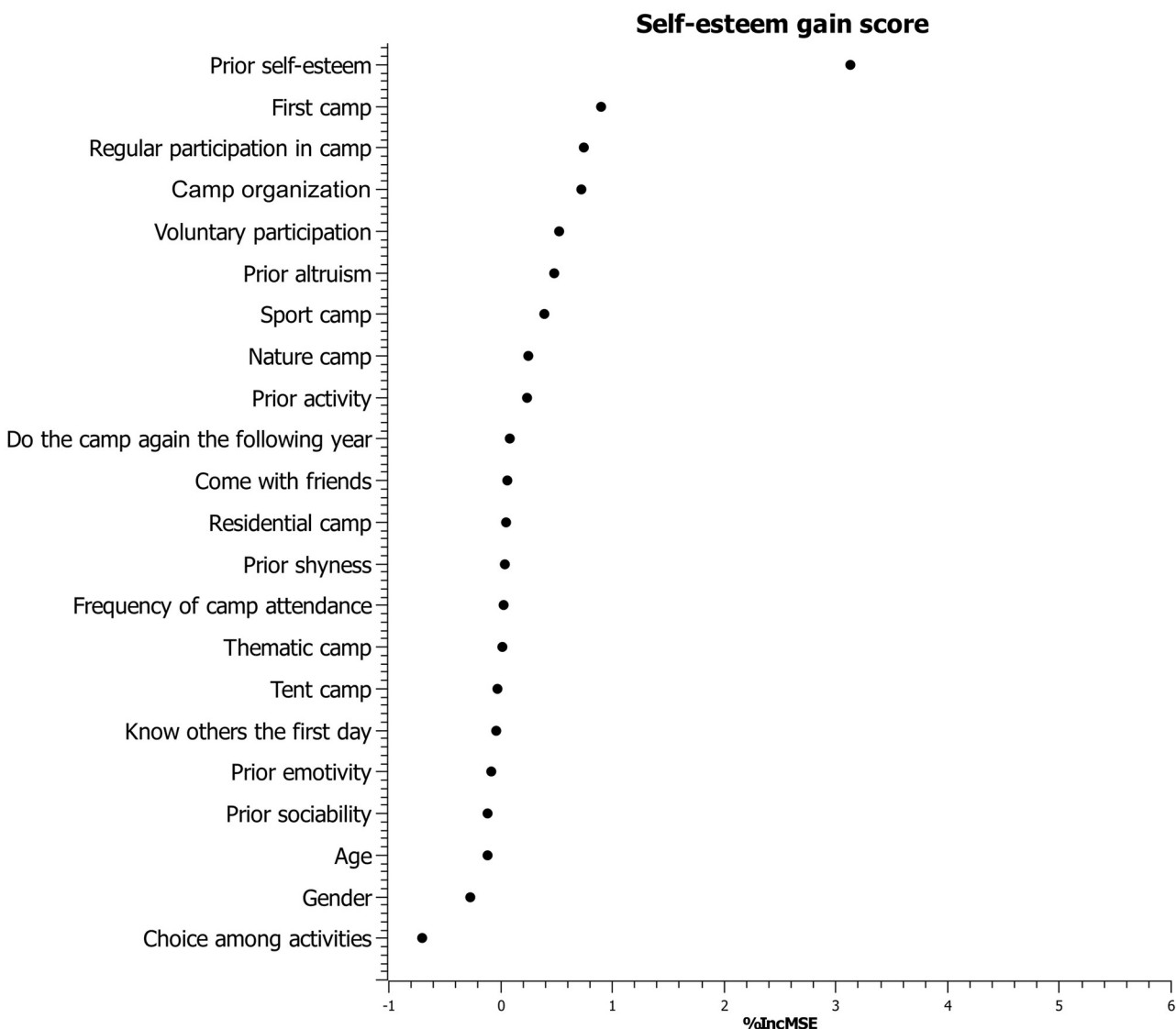

**Fig 3. Variables importance plot for self-esteem gain score in the camp condition.**

On the categorical variables, this profile is characterised by regular participation in holiday camps for all the children (100%), the desire to do the camp again the following year, also for all of them (100%) and a very high proportion of having come with friends (88.89%). On the contrary, none of these children were going to camp for the first time (0%) and less than a quarter of them did not know any other children at the start of the camp (22.22%). The results of the one-sample t-test indicate that the mean altruism gain score (M = 6.06, SD = 5.46) is significantly higher than zero ($t_{17}$ = 4.70, p < .001). There was an increase in the altruism score between the two measurement times. The effect size is strong (d = 1.11). The self-esteem gain score (M = -0.83, SD = 4.69) was not significantly different from zero for children in the first profile ($t_{17}$ = -0.75, p = .46). On average, there was no change between the two measurement times.

**Second profile.** The second profile is composed of thirty-eight children with a higher average score than the camp participants as a whole for altruism (M = 38.58, SD = 8.38) and

**Table 3. Summary of the four profiles identified by the hierarchical analysis.**

| | First profile (n = 18) | Second profile (n = 38) | Third profile (n = 20) | Fourth profile (n = 18) | Reference (n = 94) |
|---|---|---|---|---|---|
| Sociodemographic variables | | | | | |
| Age | 9.83 (1.69) | | 12.80 (1.36) | 10.11 (2.89) | 11.22 (2.22) |
| Gender | | | | | |
| boys | | | | | 46.8%. |
| girls | | | | | 53.2%. |
| Quantitative variables | | | | | |
| Altruism | 20.89 (10.39) | 38.58 (8.38) | 23.00 (11.37) | | 29.38 (12.46) |
| Self-esteem | 3.94 (4.67) | | -2.05 (5.77) | | 0.97 (4.88) |
| Emotionality | | | | | 2.64 (0.96) |
| Activity | | 3.56 (0.56) | 2.85 (0.76) | | 3.35 (0.72) |
| Sociability | | | 3.40 (1.11) | | 3.91 (0.86) |
| Shyness | | | | | 2.54 (0.76) |
| Categorical variables | | | | | |
| Participation | | | | | |
| Yes | 100% | 100% | 100% | 0% | 80.85%. |
| Frequency of camps | | | | | |
| Rarely | | | 30% | | 11.70%. |
| Often | | 31.58% | 0% | 0% | 18.09%. |
| First camp | | | | | |
| Yes | 0% | 5.26% | 5% | 94.44% | 21.28%. |
| Friends | | | | | |
| Yes | 88.89% | 73.68% | 30% | 27.78% | 58.51%. |
| Choice of activities | | | | | |
| Yes | | 100% | 55% | | 89.36%. |
| Choice of participation | | | | | |
| Yes | | | | 66.67% | 87.23%. |
| Knowing others (IR) | | | | | |
| Yes | 22.22% | | | 77.77% | 54.25%. |
| Want to do it again | | | | | |
| Yes | 100% | | 60% | | 82.97%. |

*Note*. Only those characteristics that significantly distinguish the different profiles (p < .05) are indicated in the table. The righthand column (reference) allows comparison with the average of the four profiles. The analysis was carried out with ninety-four of the one hundred and forty-five participants, because it cannot be carried out with missing data. (IR) indicates that the item is reversed. For example, 77.77% of the children in the fourth profile did not know the other participants at the beginning of the camp.

for activity (M = 3.56, SD = 0.56) on the pre-test. On the categorical variables, this profile is characterised by regular participation in holiday camps for all children (100%)—a high proportion even participate often (31.58%). All of them also indicated that they had the opportunity to make choices among the activities offered (100%). Finally, a very high proportion of them came to the camp with friends (73.68%). On the contrary, very few of these children went to camp for the first time (5.26%). The inconsistency of the percentages reflects the inconsistency of the responses. Some children responded that they were both regular participants in summer camps and that they were going to camp for the first time. The results of the one-sample t-test indicate that the average altruism gain score (M = 0.16, SD = 5.73) is not significantly different from zero ($t_{37}$ = 0.17, p = .87). On average, there was no change between the two measurement times. The mean self-esteem gain score (M = 1.13, SD = 2.41) was

significantly greater than zero ($t_{37}$ = 2.90, p = .006). There was an increase in the self-esteem score between the two measurement times. The effect size was moderate (d = 0.47).

**Third profile.** The third profile is composed of twenty children who are older than the camp participants as a whole (M = 12.80 years, SD = 1.36 years) with lower average pre-test scores for altruism (M = 23.00, SD = 11.37), self-esteem (M = -2.05, SD = 5.77), activity (M = 2.85, SD = 0.76) and sociability (M = 3.40, SD = 1.11). On the categorical variables, this profile is characterised by regular participation in holiday camps (100%)—but with low frequency (i.e. less than one camp per year) for more than a quarter of them (30%). Conversely, none of the children in the third profile indicated that they often attend holiday camps (0%). The children in this profile also stand out for their low proportion of having gone to camp for the first time (5%) and for having come with friends (30%). They are also less likely than the full camp sample to say that they had the opportunity to choose from among the activities on offer (55%) and to want to go to camp again the following year (60%). The results of the one-sample t-test indicate that the mean altruism gain score (M = 0.20, SD = 7.75) is not significantly different from zero ($t_{19}$ = 0.12, p = .91). On average, there was no change between the two measurement times. The self-esteem gain score (M = -0.65, SD = 5.52) is not significantly different from zero ($t_{19}$ = -0.53, p = .60). On average, there was no change between the two measurement times.

**Fourth profile.** The fourth profile is made up of eighteen children who are younger than all the camp participants (M = 10.11 years, SD = 2.89 years). The children in this profile do not differ on any other quantitative variable. On the categorical variables, this profile is characterised by the high proportion of children who are participating in a holiday camp for the first time (94.44%). It follows quite logically that none of the children in this profile reported participating regularly (0%). In addition, fewer of them—compared to the entire camp sample—stated that they had come to camp with friends (27.78%) and that they had voluntarily chosen to participate (66.67%). Instead, more of them indicated that they did not know any other children at the start of the camp (77.77%). The results of the one-sample t-test indicate that the mean altruism gain score (M = 1.33, SD = 4.16) is not significantly different from zero ($t_{17}$ = 1.36, p = .19). On average, there was no change between the two measurement times. The self-esteem gain score (M = -0.94, SD = 4.26) is not significantly different from zero ($t_{17}$ = -0.94, p = .36). On average, there was no change between the two measurement times.

## Discussion

This research studied the evolution of altruism and self-esteem scores in children attending summer camps. The results show a significant increase in the altruism score between the beginning and the end of the camp. The self-esteem score remained stable between the two measurement periods. Exploratory analyses revealed differences in the evolution of the children. In particular, it seems that the holiday organisation with which the children went is an important variable in the evolution of their altruism score. This is also the case for altruism, sociability and shyness scores at the beginning of the camp. The results obtained through the profiles suggest that certain factors could contribute to making the summer camp experience more favourable for the development of altruism, and self-esteem for some of them. More precisely, the hierarchical cluster analysis revealed four distinct profiles. The first profile corresponds to children who are younger than average overall, with a low initial altruism score but a high self-esteem score. These children are more familiar with summer camps. The analysis of effects showed a significant increase in altruism scores among these children after the camp, with a strong effect size. The second profile emerging from this analysis represents children who had higher than average altruism and activity scores prior to camp. They are accustomed

to camps and mostly come with their friends. They showed a significant improvement in self-esteem after the summer camp. Finally, the third profile corresponding to older children, with low scores on altruism, self-esteem, sociability and activity, and the fourth profile for younger children with no experience in summer camps, showed no significant improvement on measures of altruism and self-esteem.

In sum, according to this profile analysis, it seems that children who attend summer camps regularly, have participated with friends, and have had a positive experience, since they express the wish to repeat the camp the following year, are more likely to have had their altruism or self-esteem score increased. This is consistent with the conclusion of Bialeschki et al. [25] that summer camps are beneficial for most children but cannot provide optimal experiences for every child all the time. This result remains to be verified, however, as these factors did not emerge as important variables in predicting the difference scores for altruism and self-esteem of children who went to camp with the random forest.

Overall, the results therefore support the hypothesis of a beneficial effect of holiday camp participation on the development of altruism, which makes them congruent with the literature that notes an association between holiday camp participation and social and prosocial skills [20, 23]. On the other hand, the effect of summer camp participation on self-esteem, highlighted in other surveys [19, 23, 24], has not been replicated for the whole summer camp group. Readdick and Schaller [24] also use a multidimensional self-esteem scale, and find a significant difference between two measurement times, but only on global self-esteem and on a popularity scale. Their scale has two dimensions similar to those used in this research, one related to physical appearance and one related to intellectual abilities. On these two dimensions, in particular, they did not find any significant difference either. It is possible that some dimensions are not sufficiently relevant in a camp context to generate comparisons or judgements that could change children's self-esteem [49]. Another possibility is related to the scale itself. Maintier and Alaphilippe [35] argue that it provides an estimate of self-esteem, but children self-assess on the basis of social comparison. As a result, they may perceive themselves as inferior, equal or superior to others. However, the average self-esteem at the first measurement time is positive in both groups. This means that children are already more likely to respond that they are equal to others, or even slightly superior. Due to the possible ceiling effect, it seems difficult to detect an increase in the self-esteem score, especially in such a short time and since the effect sizes with this type of protocol are relatively small [19]. Self-esteem may indeed be less malleable during a two-week summer camp experience.

## Limits, conclusions and perspectives

One of the great challenges of summer camp studies is to evaluate their benefits precisely [50], even if this evaluation was very complex. The main limit, due to external constraints, was that the questionnaires could not be administered during the same period, the summer, for both groups: in the schools, the questionnaires were administered before and after the October holidays, whereas those of the camps were administered at the beginning and end of the stay, during the summer holidays. For future research, it would be useful to make the groups more comparable, through more systematic matching.

In conclusion, the results of the present research support the hypothesis that summer camps can contribute to the development of altruism in children and adolescents. No general trend was found in relation to the development of participants' self-esteem, but the exploratory analyses conducted suggest different patterns of development. This work shows encouraging effects and offers several interesting perspectives for future research.

## Supporting information

**S1 Data.**
(XLS)

## Acknowledgments

We would like to thank the summer camp organisations, the children and young people who participated in this study, and their parents, who helped us to collect the data.

## Author Contributions

**Conceptualization:** Edouard Gentaz, Jennifer Malsert.

**Data curation:** Yves Gerber, Jennifer Malsert.

**Formal analysis:** Yves Gerber.

**Funding acquisition:** Yves Gerber.

**Investigation:** Yves Gerber.

**Methodology:** Jennifer Malsert.

**Project administration:** Edouard Gentaz, Jennifer Malsert.

**Resources:** Edouard Gentaz, Jennifer Malsert.

**Supervision:** Edouard Gentaz, Jennifer Malsert.

**Validation:** Jennifer Malsert.

**Writing – original draft:** Yves Gerber.

**Writing – review & editing:** Edouard Gentaz, Jennifer Malsert.

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
