## [Decision Letter · Decision Letter 0]

11 Jul 2022

PONE-D-21-40215The effects of Swiss summer camp on the development of socio-emotional abilities in childrenPLOS ONE

Dear Dr. Malsert,

Thank you for submitting your manuscript to PLOS ONE. After careful consideration, we feel that it has merit but does not fully meet PLOS ONE’s publication criteria as it currently stands. Therefore, we invite you to submit a revised version of the manuscript that addresses the points raised during the review process.

Please note that we have only been able to secure a single reviewer to assess your manuscript. We are issuing a decision on your manuscript at this point to prevent further delays in the evaluation of your manuscript. Please be aware that the editor who handles your revised manuscript might find it necessary to invite additional reviewers to assess this work once the revised manuscript is submitted. However, we will aim to proceed on the basis of this single review if possible.  Your manuscript has been assessed by an expert reviewer, whose comments are appended below. The reviewer has highlighted concerns about several aspects of the methodology and framing of the research questions/hypotheses, among other issues. Please ensure you respond to each point carefully in your response to reviewers document, and modify your manuscript accordingly.

We look forward to receiving your revised manuscript.

Kind regards,

Joseph Donlan

Editorial Office

PLOS ONE

Journal Requirements:

Reviewers' comments:

Reviewer's Responses to Questions

**Comments to the Author**

1. Is the manuscript technically sound, and do the data support the conclusions?

Reviewer #1: Yes

2. Has the statistical analysis been performed appropriately and rigorously? 

Reviewer #1: Yes

3. Have the authors made all data underlying the findings in their manuscript fully available?

Reviewer #1: Yes

4. Is the manuscript presented in an intelligible fashion and written in standard English?

Reviewer #1: Yes

5. Review Comments to the Author

Reviewer #1: Dear Author(s):

I was pleased to have the opportunity to review the manuscript, “The effects of Swiss summer camp on the development of social-emotional abilities in children.” The study contributes to the body of knowledge associated with benefits of summer camp participation for youth, with a specific focus on altruism and self-esteem as targeted outcomes.

Provided below are my recommended edits, comments, and questions.

Page 3, line 44

• Please clarify—“…target for universal prevention” of what?

Page 3, line 49

• Replace “these” with “organised activities”

Page 3, line 54

• Should “spells” be “sports”

Page 4, line 61

• Recommend replacing “several adults” with “supportive adults”

Page 4, line 68

• Shouldn’t “This is the first” be “This was the first”?

Page 5, line 84

• Please define “holiday camps” for international readers who may be unfamiliar with the term.

Page 5

• General comment: Adding specific hypotheses or research questions after the purpose (aim) statements would improve the manuscript.

Page 6

• General comment: Identifying the study as having a quasi-experimental design is recommended, as many studies of summer camp have used cross sectional designs.

Pages 6-8

• General comment: Good use of previously designed measures that were also validated through Confirmatory Factor Analysis.

Page 11

• General question: With regard to the data preparation and analyses, how were outliers, the distribution of the data, and missing data analyzed/treated? Please add this detail.

Page 12

• General comment: I appreciated the author(s) decision to use hierarchical clustering, an approach that is not often used with the summer camp literature. Such cluster analysis has the potential to better inform characteristics of youth that may be more likely to benefit from summer camp experiences.

Page 13, line 271

• Recommend replacing “more than” with “when compared to”

Page 14, line 292

• General comment: Some scholarly literature suggests it’s not necessary to report effect sizes for non-significant differences. If you decided to include effect sizes in this case, you might explain why such information is important for readers to consider.

Page 15, line 308

• I believe “camp organism” should read “camp organization”

Page 15, lines 312-313

• The figures are not labelled in the manuscript I reviewed.

Pages 15-19

• General comment: It’s not clear why the profiles are included in the Results section, because they aren’t discussed at all. It seems entirely unnecessary. As noted earlier in my review, I think these analyses are very interesting and can make a valuable contribution to the summer camp literature. However, these profiles need attention to the Discussion, otherwise they should be dropped from the paper.

Page 20, lines 413-420

• The flow of this portion of the paragraph is a bit awkward. I recommend moving the sentence that begins, “This is consistent with…” so that it comes after the sentence that begins, “According to the profile analysis…”

Page 21, lines 434-438

• General comment: Self-esteem may be more stable and less malleable through a 1-2 week summer camp experience.

Page 21, lines 441-443

• General comment: You might point out the influence of a possible “ceiling effect” here, which also limits variance in your data.

Page 21, line 446

• Recommend moving the word “precisely” so it comes after “benefits” in line 447.

6. PLOS authors have the option to publish the peer review history of their article (what does this mean?). If published, this will include your full peer review and any attached files.

Reviewer #1: No

---

## [Author Response · Author response to Decision Letter 0]

17 Jul 2022

Points raised by the reviewer and responses: 

• Page 3, line 44

Please clarify —“…target for universal prevention” of what?

We added clarification: “(e.g. school bullying, school failure, youth violence)”.

• Page 3, line 49

Replace “these” with “organised activities”

Done.

• Page 3, line 54

Should “spells” be “sports”

Done.

• Page 4, line 61

Recommend replacing “several adults” with “supportive adults”

Done.

• Page 4, line 68

Shouldn’t “This is the first” be “This was the first”?

Done.

• Page 5, line 84

Please define “holiday camps” for international readers who may be unfamiliar with the term.

Done: “defined as residential and structured extracurricular activities taking place over several days during the holiday periods”.

• Page 5

General comment: Adding specific hypotheses or research questions after the purpose (aim) statements would improve the manuscript.

Two research questions were added: 

“The two research questions of the study were: (1) To what extent participation in summer camps is associated with increased self-esteem and altruism in children and adolescents aged 6–16 years? (2) What factors may influence the development of self-esteem and altruism in children and adolescents aged 6 to 16 who attend summer camps?”

• Page 6

General comment: Identifying the study as having a quasi-experimental design is recommended, as many studies of summer camp have used cross sectional designs.

We added quasi-experimental in abstract and “took part in this quasi-experimental design study” in method section.

• Pages 6-8

General comment: Good use of previously designed measures that were also validated through Confirmatory Factor Analysis.

Thank you for this comment. 

• Page 11

General question: With regard to the data preparation and analyses, how were outliers, the distribution of the data, and missing data analyzed/treated? Please add this detail.

We added: 

“The assumptions of the statistical analyses were checked. In case of outliers or violation of the normality assumption, the nonparametric alternatives proposed by Broc et al. (2016) were also performed. In this article, the results of the parametric analyses, which are relatively robust (e.g. Cessie et al., 2020; Knief & Forstmeier, 2021) and lead to the same conclusions, are reported. Missing data were excluded analysis by analysis.”

• Page 12

General comment: I appreciated the author(s) decision to use hierarchical clustering, an approach that is not often used with the summer camp literature. Such cluster analysis has the potential to better inform characteristics of youth that may be more likely to benefit from summer camp experiences.

Thank you, it seems important for us for a better understanding of this effects. 

• Page 13, line 271

Recommend replacing “more than” with “when compared to”

Done.

• Page 14, line 292

General comment: Some scholarly literature suggests it’s not necessary to report effect sizes for non-significant differences. If you decided to include effect sizes in this case, you might explain why such information is important for readers to consider.

Indeed, we deleted the effect sizes for non-significant values. 

• Page 15, line 308

I believe “camp organism” should read “camp organization”

Done (line 317).

• Page 15, lines 312-313

The figures are not labelled in the manuscript I reviewed.

We verified and added the missing elements.

• Pages 15-19

General comment: It’s not clear why the profiles are included in the Results section, because they aren’t discussed at all. It seems entirely unnecessary. As noted earlier in my review, I think these analyses are very interesting and can make a valuable contribution to the summer camp literature. However, these profiles need attention to the Discussion, otherwise they should be dropped from the paper.

Indeed, a point of discussion on these different profiles was missing to enhance the interest of these analyses. We added a whole paragraph in the discussion to summarize the main observations made by the profile analysis (Lines 422-437).

• Page 20, lines 413-420

The flow of this portion of the paragraph is a bit awkward. I recommend moving the sentence that begins, “This is consistent with…” so that it comes after the sentence that begins, “According to the profile analysis…”

The sentence was moved (lines 437-439).

• Page 21, lines 434-438

General comment: Self-esteem may be more stable and less malleable through a 1-2 week summer camp experience.

Indeed, we added “Self-esteem may indeed be less malleable during a two-week summer camp experience” (Line 463).

• Page 21, lines 441-443

General comment: You might point out the influence of a possible “ceiling effect” here, which also limits variance in your data.

Effectively, we added “Due to the possible ceiling effect, it seems difficult to detect an increase in the self-esteem score” (Lines 460-461).

• Page 21, line 446

Recommend moving the word “precisely” so it comes after “benefits” in line 447.

Done (Lines 466-467).

---

## [Decision Letter · Decision Letter 1]

12 Oct 2022

The effects of Swiss summer camp on the development of socio-emotional abilities in children

PONE-D-21-40215R1

Dear Dr. Malsert,

We’re pleased to inform you that your manuscript has been judged scientifically suitable for publication and will be formally accepted for publication once it meets all outstanding technical requirements.

Kind regards,

Shazia Khalid, PhD

Academic Editor

PLOS ONE

Additional Editor Comments (optional):

Reviewers' comments:

Reviewer's Responses to Questions

**Comments to the Author**

1. If the authors have adequately addressed your comments raised in a previous round of review and you feel that this manuscript is now acceptable for publication, you may indicate that here to bypass the “Comments to the Author” section, enter your conflict of interest statement in the “Confidential to Editor” section, and submit your "Accept" recommendation.

Reviewer #1: All comments have been addressed

2. Is the manuscript technically sound, and do the data support the conclusions?

Reviewer #1: Yes

3. Has the statistical analysis been performed appropriately and rigorously? 

Reviewer #1: Yes

4. Have the authors made all data underlying the findings in their manuscript fully available?

Reviewer #1: Yes

5. Is the manuscript presented in an intelligible fashion and written in standard English?

Reviewer #1: Yes

6. Review Comments to the Author

Reviewer #1: Thank you for addressing my questions, comments, and concerns in your revisions. The manuscript is much improved. Nicely done!

7. PLOS authors have the option to publish the peer review history of their article (what does this mean?). If published, this will include your full peer review and any attached files.

Reviewer #1: No

---

## [Editor Report · Acceptance letter]

18 Oct 2022

PONE-D-21-40215R1 

The effects of Swiss summer camp on the development of socio-emotional abilities in children 

Dear Dr. Malsert:

I'm pleased to inform you that your manuscript has been deemed suitable for publication in PLOS ONE. Congratulations! Your manuscript is now with our production department. 

Kind regards, 

on behalf of

Professor Shazia Khalid 

Academic Editor

PLOS ONE